# Effects of Evolocumab on Carotid Intima-Media Thickness and Clinical Parameters in Patients Taking a Statin

**DOI:** 10.3390/jcm9072256

**Published:** 2020-07-16

**Authors:** Keiji Hirai, Shigeki Imamura, Aizan Hirai, Susumu Ookawara, Yoshiyuki Morishita

**Affiliations:** 1Division of Nephrology, First Department of Integrated Medicine, Saitama Medical Center, Jichi Medical University, 1-847 Amanuma-cho, Omiya-ku, Saitama-shi, Saitama-ken 330-8503, Japan; su-ooka@hb.tp1.jp (S.O.); ymori@jichi.ac.jp (Y.M.); 2Department of Internal Medicine, Chiba Cerebral and Cardiovascular Center, 575 Tsurumai, Ichihara-shi, Chiba-ken 290-0512, Japan; shigeimam@gmail.com (S.I.); aizanvaio4909@gmail.com (A.H.)

**Keywords:** atherosclerosis, carotid intima-media thickness, evolocumab, proprotein convertase subtilisin/kexin type 9

## Abstract

We determined the effects of evolocumab, a fully human monoclonal antibody targeting proprotein convertase subtilisin/kexin type 9, on carotid intima-media thickness (IMT) and the factors associated with the change in carotid IMT in patients taking a statin. The change in carotid mean and maximum IMT before and after the initiation of evolocumab treatment was retrospectively analyzed in 229 statin-treated patients. The changes in clinical parameters, including serum lipid concentrations, were also evaluated. Evolocumab significantly reduced the increase in carotid mean and maximum IMT (0.09 ± 0.13 mm/year to −0.04 ± 0.16 mm/year, *p* < 0.001 and 0.17 ± 0.38 mm/year to 0.08 ± 0.47 mm/year, *p* = 0.02). Evolocumab reduced serum total cholesterol, low-density lipoprotein-cholesterol, triglyceride, and lipoprotein (a) concentrations (each *p* < 0.001), and increased serum high-density lipoprotein (HDL)-cholesterol concentrations (*p* = 0.01). Multiple linear regression analysis revealed that the change in HDL-cholesterol (standard coefficient (*β*) = −0.120, *p* = 0.04) and carotid mean IMT (*β* = −0.467, *p* < 0.001) were independently correlated with the change in carotid mean IMT during the administration of evolocumab, whereas the change in HDL-cholesterol (*β* = −0.208, *p* = 0.002) and log-triglyceride (*β* = −0.167, *p* = 0.01) independently correlated with the change in carotid maximum IMT. Evolocumab reduced the increase in carotid IMT in patients taking a statin. These results suggest that evolocumab is protective against carotid atherosclerosis in patients undergoing statin therapy.

## 1. Introduction

Atherosclerosis is a chronic vascular inflammatory disease that involves the arterial wall and is a common cause of cardiovascular diseases, including coronary artery disease and stroke [1]. The prevalence of these diseases is still increasing and they are the leading causes of death worldwide [2]. Therefore, strategies aimed at slowing the progression of atherosclerosis are important for the prevention of coronary artery disease and stroke.

Carotid intima-media thickness (IMT) is a surrogate marker of atherosclerotic disease [3]. The measurement of carotid IMT using ultrasonography is easy, noninvasive, and repeatable. In addition, several studies have reported that an increase in carotid IMT is associated with a higher risk of atherosclerotic sequelae, including coronary artery disease and stroke [4,5,6].

Evolocumab is a fully human monoclonal antibody that targets proprotein convertase subtilisin/kexin type 9 (PCSK9) and lowers the circulating low-density lipoprotein (LDL)-cholesterol concentration by 60–70% in patients who are at high risk of cardiovascular events and who are undergoing statin therapy [7,8]. A recent randomized controlled trial showed that evolocumab reduced the progression of coronary atherosclerosis in statin-treated patients [9]. However, it remains uncertain whether evolocumab can reduce the progression of carotid atherosclerosis in patients taking statins. Furthermore, factors associated with the regression of carotid atherosclerosis under evolocumab treatment has not yet been determined. Therefore, in the present study, we determined the effects of evolocumab on carotid IMT and the factors associated with the change in carotid IMT, in patients taking a statin.

## 2. Patients and Methods

### 2.1. Ethical Approval

This study was performed in accordance with the ethical principles contained in the Declaration of Helsinki and was approved by the Ethics Committee of the Chiba Cerebral and Cardiovascular Center (J-480). Informed consent was not required by the Ethics Committee because of the retrospective design of the study. Patient data were stored carefully to maintain their confidentiality. Information regarding this study, including the patients’ right to opt out, has been uploaded to the website of the Chiba Cerebral and Cardiovascular Center.

### 2.2. Patients

In our institute, most patients undergo carotid artery ultrasonography every 1–2 years to identify those at high risk of developing coronary artery disease [10]. Intensive lipid-lowering therapies, including the use of statins and PCSK9 inhibitors, are administered to high-risk patients to prevent the development of acute coronary syndrome [10]. In Japan, PCSK9 inhibitors have been indicated for the treatment of patients with familial hypercholesterolemia or hypercholesterolemia who have a high risk for cardiovascular events and do not adequately respond to statins. Therefore, evolocumab was administered to hypercholesterolemic patients with confirmed cardiovascular risk factors who did not achieve optimal LDL-cholesterol goals or exhibited progression of carotid IMT despite statin therapy. We retrospectively analyzed the clinical and laboratory data of patients who had regularly visited the Chiba Cerebral and Cardiovascular Center between April 2016 and March 2019.

The inclusion criteria were—(i) age ≥20 years, (ii) treatment with evolocumab for ≥12 months, (iii), carotid artery ultrasonography at the initiation of evolocumab, as well as 12 months (±1 month) before and 12 months (±1 month) afterwards, (iv) treatment with a statin for ≥12 months before the initiation of evolocumab. The exclusion criterion was renal replacement therapy.

### 2.3. Study Design

Figure 1 shows the study design. The study was a retrospective observational study of 229 patients. Demographic and clinical data were obtained by retrospective review of the patients’ medical records. Evolocumab was administered subcutaneously at a dose of 140 mg every 2 weeks on the same day of the week. The change in carotid mean and maximum IMT was compared between the 12 months before and the 12 months after the initiation of evolocumab treatment. The factors that were independently associated with the change in carotid mean and maximum IMT during the administration of evolocumab were analyzed using multiple linear regression analysis. Serum lipid concentrations (total cholesterol, LDL-cholesterol, high-density lipoprotein (HDL)-cholesterol, triglyceride, and lipoprotein (a)), were also measured at the initiation and after 12 months of evolocumab administration.

### 2.4. Laboratory Methods

Blood and urine parameters were determined in the Department of Clinical Laboratory, Chiba Cerebral and Cardiovascular Center. Serum hemoglobin A1c (HbA1c) levels are shown as National Glycohemoglobin Standardization Program values. The estimated glomerular filtration rate (eGFR) was calculated using a modified version of the Modification of Diet in Renal Disease formula of the Japanese Society of Nephrology, as follows—eGFR (mL/min/1.73 m^2^) = 194 × age^−0.287^ × serum creatinine^−1.094^ (multiplied by 0.739 for women) [11]. Hypertension was defined by a systolic blood pressure ≥140 mmHg and/or a diastolic blood pressure ≥90 mmHg, or the current use of antihypertensive agents. Diabetes mellitus was defined as an HbA1c level ≥6.5% or the use of oral hypoglycemic agents and/or insulin therapy.

### 2.5. Ultrasonographic Measurement of Carotid IMT

Carotid IMT was measured using B-mode ultrasound imaging with a 7.5-MHz linear transducer (Aplio MX; Toshiba Medical Systems, Tokyo, Japan). Carotid IMT was measured as the distance between two parallel echogenic lines representing the lumen-intima interface and the media-adventitia interface on the posterior wall of the artery (Figure 2A) [12]. The carotid mean IMT was defined as the average of all mean IMT values obtained from the left and right common carotid artery, carotid bulb, and internal carotid artery, which were determined by using an automated edge-detection system [13]. The carotid maximum IMT was defined as the highest carotid IMT measured manually on both sides of the common carotid artery, carotid bulb, and internal carotid artery [13] (Figure 2B). All the scans were performed by experienced laboratory technicians.

### 2.6. Statistics

Statistical analysis was performed using JMP 11 (SAS Institute, Inc., Cary, NC, USA). Continuous variables were expressed as mean ± standard deviation for a normal distribution and as median (interquartile range) for a non-normal distribution. Categorical variables were expressed as numbers and percentages. Triglycerides, lipoprotein (a), and urine albumin/creatinine ratio were not normally distributed; therefore, these valuables were transformed using a natural logarithm. The comparison of the change in carotid mean and maximum IMT in the 12 months before and after the initiation of evolocumab was performed using a paired *t*-test. Parameters that appeared to be significantly correlated with the annual change in carotid mean and maximum IMT during the administration of evolocumab in simple linear regression analyses (*p* < 0.10) were included in the multiple linear regression analysis to identify those that were independently related to the change in carotid mean and maximum IMT during the administration of evolocumab. Comparisons of laboratory data before and after the administration of evolocumab were performed using the paired t-test for normally distributed data (total cholesterol, LDL-cholesterol, HDL-cholesterol, uric acid, HbA1c, and eGFR) and the Wilcoxon signed-rank test for non-normally distributed data (triglycerides, lipoprotein (a), urine albumin/creatinine ratio, alanine aminotransferase, and creatine phosphokinase). *p* < 0.05 was considered to represent statistical significance.

## 3. Results

### 3.1. Patient Characteristics

The baseline characteristics of the patients and their medication are summarized in Table 1. The flow diagram for the participants is shown as Figure 3. Data from a total of 229 patients (148 men and 81 women; mean age: 72.6 ± 8.6 years) were analyzed. Their carotid mean and maximum IMT at the initiation of evolocumab was 1.3 ± 0.3 and 2.5 ± 0.7 mm, respectively. All the patients were taking a statin and 46 (20.1%) had a history of coronary artery disease. The percentages of the participants with hypertension, diabetes mellitus, and familial hypercholesterolemia were 76.9%, 55.9%, and 0.9%, respectively. The doses of each statin administered are summarized in Table 2.

### 3.2. Effects of Evolocumab on Carotid Mean and Maximum IMT, Assessed Ultrasonographically

The changes in carotid mean IMT before and after the administration of evolocumab are shown in Figure 4. The change in carotid mean IMT improved significantly from 0.09 ± 0.13 mm/year before the initiation of evolocumab to −0.04 ± 0.16 mm/year afterwards (*p* < 0.001) (Figure 5). The changes in carotid maximum IMT before and after the administration of evolocumab are shown in Figure 6. The change in carotid maximum IMT also improved significantly from 0.17 ± 0.38 mm/year before the initiation of evolocumab to 0.08 ± 0.47 mm/year afterwards (*p* = 0.02) (Figure 7).

### 3.3. Factors Associated with the Change in Carotid Mean and Maximum IMT During the Administration of Evolocumab

Simple linear regression analyses revealed that the change in carotid mean IMT during the administration of evolocumab significantly correlated with the change in eGFR and the baseline carotid mean IMT (Table 3), whereas the change in carotid maximum IMT significantly correlated with the change in serum HDL-cholesterol concentration, the use of an antiplatelet agent, and the use of a renin-angiotensin system inhibitor (Table 4). We then performed a multiple linear regression analysis using the variables that were marginally or statistically significantly correlated (*p* < 0.10) with the change in carotid mean and maximum IMT in the simple linear regression analyses. This revealed that the change in HDL-cholesterol (standard coefficient (*β*) = −0.120, *p* = 0.04) and the baseline carotid mean IMT (*β* = −0.467, *p* < 0.001) independently correlated with the change in carotid mean IMT during the administration of evolocumab, whereas the change in HDL-cholesterol (*β* = −0.208, *p* = 0.002) and log-triglycerides (*β* = −0.167, *p* = 0.01) independently correlated with the change in carotid maximum IMT.

### 3.4. Changes in Serum Lipid Concentrations

The serum total cholesterol, LDL-cholesterol, triglyceride, and lipoprotein (a) concentrations significantly decreased from 149.1 ± 31.7, 69.4 ± 24.1, 107 (83–151), and 15 (4–30) mg/dL at the initiation of evolocumab to 94.3 ± 25.5, 20.8 ± 16.8, 90 (63–125), and 6 (2–17) mg/dL after 12 months, respectively (each *p* < 0.001). In contrast, the HDL-cholesterol concentration significantly increased from 53.9 ± 14.0 to 55.4 ± 15.0 mg/dL after 12 months of treatment (*p* = 0.01) (Table 5).

### 3.5. Changes in Other Laboratory Parameters and Adverse Effects

There were no changes in serum uric acid concentration, urine albumin/creatinine ratio, serum alanine aminotransferase activity, or serum creatine phosphokinase activity between the initiation of evolocumab and 12 months afterwards (Table 5). The HbA1c concentration significantly increased from 6.3 ± 0.9% to 6.5 ± 1.1% after 12 months of evolocumab (*p* = 0.007), whereas the eGFR significantly decreased from 68.1 ± 17.1 mL/min/1.73 m^2^ to 66.1 ± 16.6 mL/min/1.73 m^2^ after 12 months of evolocumab (*p* < 0.001) (Table 5). Adverse effects were observed in two participants—general fatigue in one and leg cramps in the other. However, they were tolerant of evolocumab and its administration was continued.

## 4. Discussion

In the present study, we have determined the effects of evolocumab on carotid IMT and the factors associated with the change in carotid IMT, in patients taking a statin. Evolocumab ameliorated the progression of carotid mean and maximum IMT in patients undergoing statin therapy, without any inducing serious adverse effects. It also had a favorable effect on serum lipid profile. Multiple linear regression analysis revealed that the change in the serum HDL-cholesterol concentration and the baseline carotid mean IMT independently correlated with the change in carotid mean IMT during the administration of evolocumab, whereas the change in serum HDL-cholesterol and triglyceride concentration independently correlated with the change in carotid maximum IMT. These results suggest that evolocumab protects against carotid atherosclerosis, including by increasing the concentration of HDL-cholesterol and decreasing the concentration of triglycerides, in patients undergoing statin therapy.

Carotid IMT is a well-established surrogate marker of atherosclerosis [3], and is associated with various atherosclerotic risk factors, such as hypertension, diabetes mellitus, and dyslipidemia [14]. A previous study has reported that baseline carotid mean IMT thickness was negatively correlated with the change in carotid mean IMT [15], which is compatible with the findings of our study. Atherosclerosis is a chronic vascular inflammatory disease in which lipid-loaded and activated macrophages play a pivotal role [16]. Injury to the vascular endothelium by various mechanisms, including high blood pressure, hyperglycemia, and dyslipidemia, leads to the infiltration and retention of monocytes in the subendothelial space [16]. These monocytes differentiate into macrophages, engulf oxidized LDL-cholesterol through macrophage scavenger receptors, and finally transform into foam cells [16]. This process induces intimal thickening and the formation of a lipid core that contains lipid-loaded foam cells covered with a layer of connective tissue [16].

PCSK9 is a proprotein convertase that is produced in hepatocytes, vascular smooth muscle cells, endothelial cells, and macrophages [17]. It increases the clearance of LDL-cholesterol receptors by hepatocytes by facilitating their lysosomal degradation, which contributes to an increase in serum LDL-cholesterol concentration [18]. Several studies have shown that evolocumab reduced LDL-cholesterol concentration by 60–70%, as well as reducing triglyceride and lipoprotein (a) concentrations and increasing HDL-cholesterol concentration [7,8]. The results of these previous studies are consistent with the findings of the present study. PCSK9 also inhibits adenosine triphosphate (ATP)-binding cassette transporter A1 expression in macrophages, secondary to LDL receptor depletion, thereby reducing cholesterol efflux from lipid-loaded foam cells [19]. Evolocumab has been reported to increase the uptake of serum LDL-cholesterol by hepatocytes through the inhibition of LDL-cholesterol receptor clearance [20] and to increase ATP-binding cassette transporter A1 expression in macrophages by upregulating LDL receptor expression [19]. These findings suggest that evolocumab attenuates the progression of carotid atherosclerosis through a reduction in serum LDL-cholesterol concentration and an increase in cholesterol efflux from lipid-loaded foam cells.

However, in the present study, neither LDL-cholesterol, nor the change in LDL-cholesterol concentration, correlated with the change in carotid IMT during the administration of evolocumab. It has been reported that there are no associations between serum LDL-cholesterol, the change in LDL-cholesterol, and the change in coronary atherosclerotic plaque size in patients who achieve an LDL-cholesterol concentration <70 mg/dL while on statin therapy [21]. In the present study, the mean LDL-cholesterol concentration at baseline was well-controlled by statin therapy (69.4 ± 24.1 mg/dL). This may explain why neither serum LDL-cholesterol nor the change in LDL-cholesterol concentration were associated with the change in carotid IMT during the administration of evolocumab in the present study.

HDL is a lipoprotein that is principally composed of phospholipids and apolipoproteins (ApoA-I, ApoA-II, and ApoC) and promotes cellular cholesterol efflux and reverse cholesterol transport [22]. HDL collects free cholesterol from cell surfaces via the ATP-binding cassette transporter and transports it to the liver for excretion in the bile [22]. Statins increase the synthesis of ApoA-Ⅰ and HDL in the liver and increase the expression of the ATP-binding cassette transporter in peripheral cells [23]. Similarly, evolocumab increases the circulating concentrations of HDL and ApoA-Ⅰ [20] and increases the expression of ATP-binding cassette transporter in macrophages [19]. It has been reported that the change in serum HDL-cholesterol concentration negatively correlated with the change in carotid IMT in patients taking statins [24]. In the present study, the change in HDL-cholesterol concentration negatively correlated with the change in carotid mean and maximum IMT during the administration of evolocumab in patients who were already taking statins. These findings would tend to support the premise that reverse cholesterol transport was most likely activated due to the use of evolocumab in our study, but an increase in the rate of reverse cholesterol transport was not specifically quantified in the present study.

Triglycerides are the major component of triglyceride-rich lipoproteins such as chylomicron and very low-density lipoprotein. In circulating blood, chylomicrons and very low-density lipoprotein are catabolized rapidly by lipoprotein lipase on blood vessel walls, producing chylomicron remnants and very low-density lipoprotein remnants [25]. Chylomicron remnants and very low-density lipoprotein remnants can directly penetrate the arterial intima and be taken up by macrophages without undergoing oxidative modifications observed with LDL particles [26]. It has been reported that the change in serum triglyceride concentration positively correlated with the change in carotid IMT in patients taking statins [24]. Several studies have shown that evolocumab reduced triglyceride concentration by 15–30%, alongside decreases in LDL-cholesterol concentration and an increase in HDL-cholesterol [7,8], which are consistent with the findings of our study. In the present study, the change in triglyceride concentration positively correlated with the change in carotid maximum IMT during the administration of evolocumab in patients who were already taking statins. These results suggest that the effects of evolocumab on carotid IMT might be partially explained by the reduction in triglycerides. Further studies are required to clarify the factors mediating the effect of evolocumab on carotid atherosclerosis in patients undergoing statin therapy.

This is the first study to show the beneficial effects of evolocumab on carotid IMT in patients undergoing statin therapy. An observational study reported that carotid maximum IMT was 2.2 ± 0.8 mm and the change in carotid maximum IMT was −0.012 mm/year in patients with coronary artery disease and carotid plaques [4]. In the present study, baseline carotid maximum IMT was 2.5 ± 0.7 mm and the change in carotid maximum IMT before the initiation of evolocumab was 0.17 mm/year. Therefore, the patients in the present study had more advanced carotid atherosclerosis and were at high risk of cardiovascular diseases. It has been reported that lipid-lowering therapy with atorvastatin did not reduce the increase in carotid maximum IMT in patients with carotid atherosclerotic plaques [27]. In the present study, evolocumab reduced the increase in carotid maximum IMT in patients taking a statin. These results suggest that evolocumab attenuates the progression of carotid atherosclerosis in patients with advanced carotid atherosclerosis and at high risk of cardiovascular diseases. In the present study, a mild increase in HbA1c concentration was observed. An animal study reported that PCSK9-knockout mice exhibited decreased insulin secretion and increased glucose intolerance [28]. However, a recent randomized control trial showed that evolocumab did not affect HbA1c level, fasting glucose level, and insulin resistance [29]. Further studies are needed to investigate the effect of evolocumab on glucose metabolism in patients taking statins.

The present study had several limitations. First, it was a retrospective observational study that might have been subject to patient selection bias. Second, it featured a before-after study design, with no control group. Therefore, we cannot exclude the possibility that other drugs, including the statins, may have affected the study results. Third, all the study participants were recruited at a single center, which might restrict the generalizability of our findings. Fourth, the percentage of familial hypercholesteremia was very low (0.9%) in this study, which influenced the results of this study because the proportion of familial hypercholesterolemia was higher in other clinical studies [7,8]. Fifth, the manual measurement of carotid maximum IMT might have caused variation and affected the results of this study. The inter- and intrasonographer reproducibility was not assessed in this study. Therefore, further multicenter, prospective studies with an appropriate control group are required to confirm the efficacy of evolocumab for the attenuation of carotid atherosclerosis progression.

In conclusion, evolocumab attenuates the progression of carotid mean and maximum IMT in patients on statin therapy, without causing any serious adverse effects. Serum HDL-cholesterol, triglycerides, and carotid IMT were associated with the change in carotid IMT during the administration of evolocumab. These results suggest that evolocumab protects against carotid atherosclerosis in these patients.

## Figures and Tables

**Figure 1 jcm-09-02256-f001:**
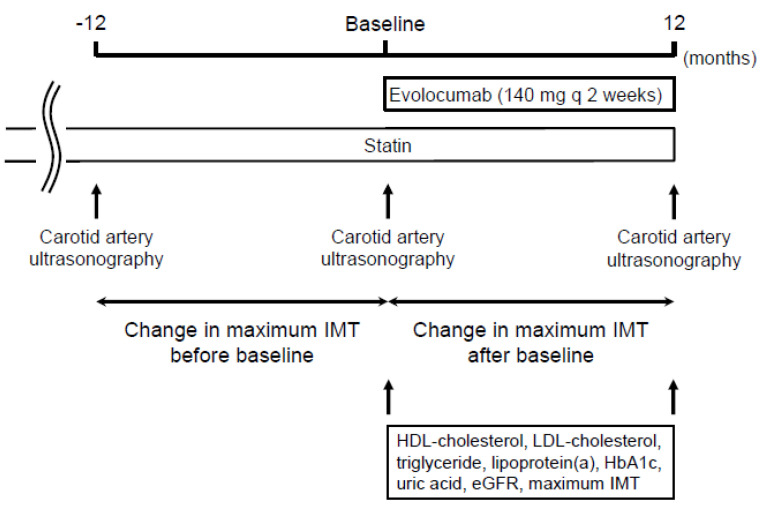
Study design.

**Figure 2 jcm-09-02256-f002:**
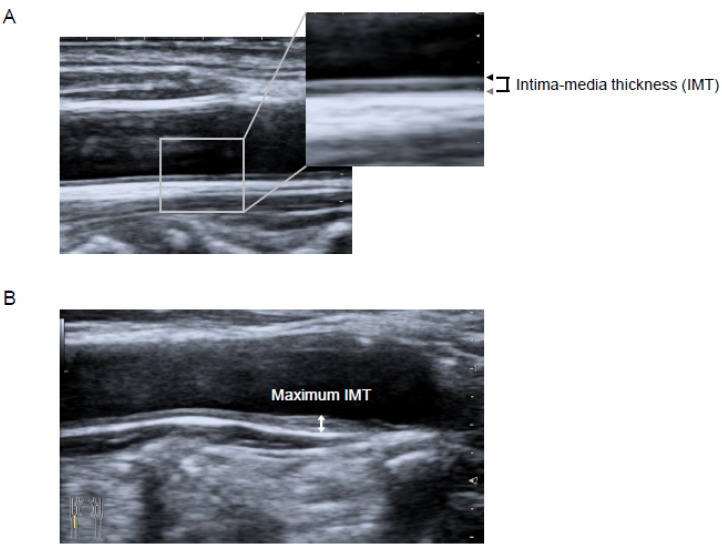
Longitudinal B-mode ultrasonographic images of the common carotid artery. (**A**) Intima-media thickness (IMT) was measured as the distance between the lumen-intima interface (black arrow) and the media-adventitia interface (gray arrow). (**B**) The maximum IMT was recorded as the largest IMT value measured on both sides of the common carotid artery, carotid bulb, and internal carotid artery (two-headed arrow). Abbreviation—IMT, intima-media thickness.

**Figure 3 jcm-09-02256-f003:**
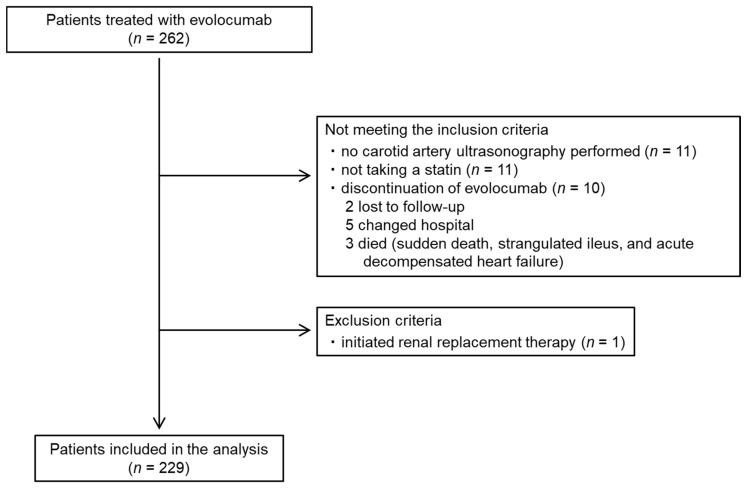
Participant flow diagram.

**Figure 4 jcm-09-02256-f004:**
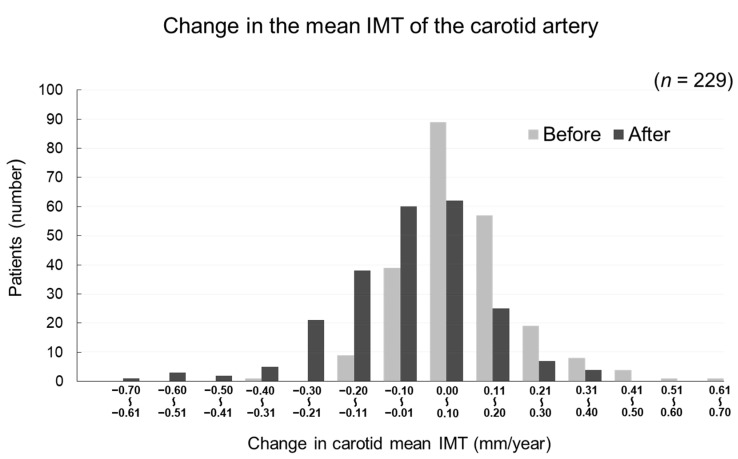
Distribution of participants according to the change in carotid mean IMT during the 12 months before and after the administration of evolocumab.

**Figure 5 jcm-09-02256-f005:**
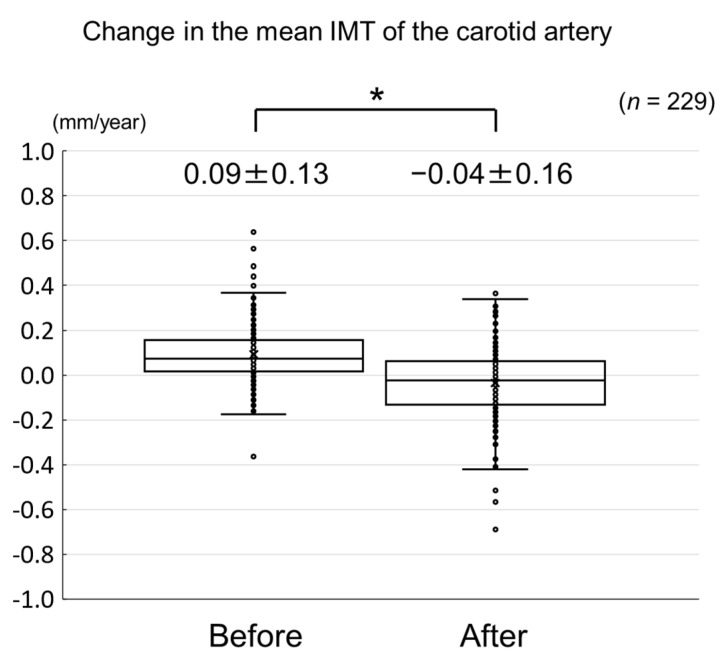
Change in carotid mean IMT during the 12 months before and after the initiation of evolocumab treatment. * *p* < 0.001.

**Figure 6 jcm-09-02256-f006:**
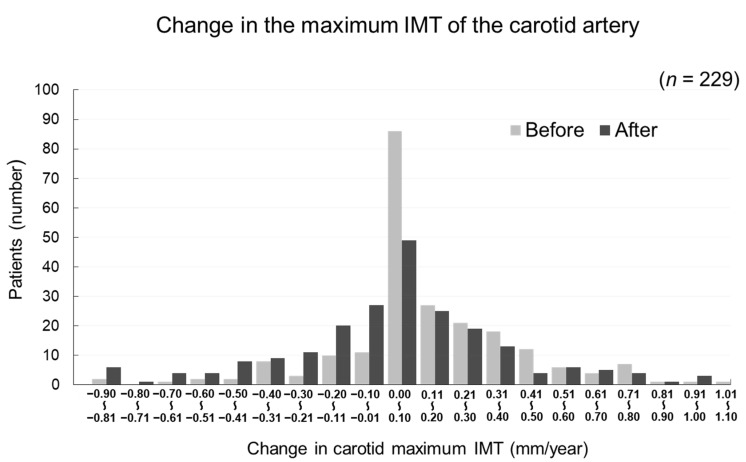
Distribution of participants according to the change in carotid maximum IMT during the 12 months before and after the administration of evolocumab.

**Figure 7 jcm-09-02256-f007:**
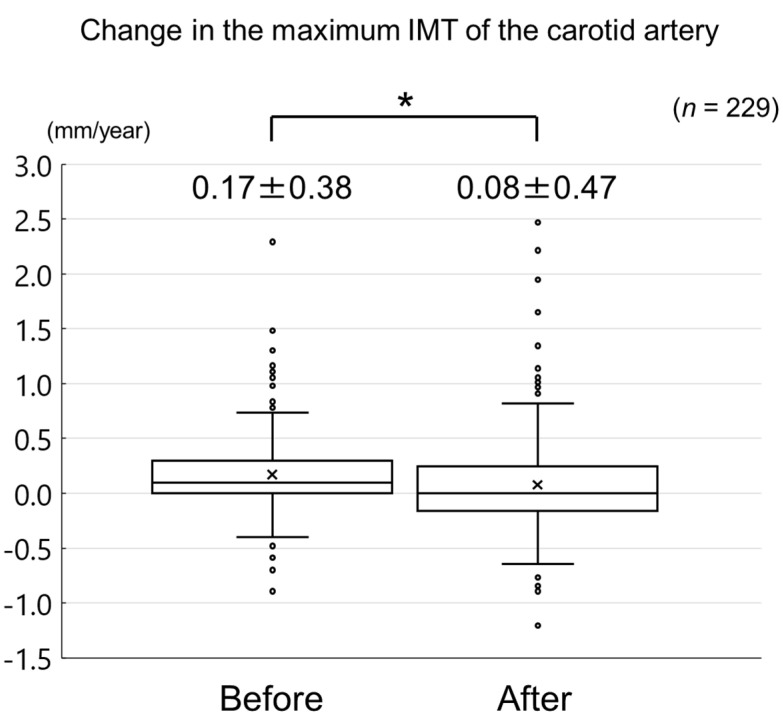
Change in carotid maximum IMT during the 12 months before and after the initiation of evolocumab treatment. * *p* = 0.02.

**Table 1 jcm-09-02256-t001:** Participant characteristics and medication at the initiation of evolocumab.

	All Participants (*n* = 229)
Age (years)	72.6 ± 8.6
Male sex (number, %)	148 (64.6)
Body mass index (kg/m^2^)	24.7 ± 3.7
Systolic blood pressure (mmHg)	133.2 ± 16.9
Diastolic blood pressure (mmHg)	76.5 ± 10.8
Hypertension (number, %)	176 (76.9)
Diabetes mellitus (number, %)	128 (55.9)
Familial hypercholesterolemia (number, %)	2 (0.9)
Coronary artery disease (number, %)	46 (20.1)
Previous myocardial infarction (number, %)	21 (9.2)
Past or current smoker (number, %)	129 (56.3)
Statin (number, %)	229 (100.0)
Ezetimibe (number, %)	25 (10.9)
Probucol (number, %)	10 (4.4)
Eicosapentaenoic acid (number, %)	195 (85.2)
Antiplatelet agent (number, %)	102 (44.5)
Renin-angiotensin system inhibitor (number, %)	112 (48.9)
*β*-blocker (number, %)	28 (12.2)
Calcium channel blocker (number, %)	126 (55.0)
Diuretic (number, %)	7 (3.1)
Anti-hyperuricemic drug (number, %)	99 (43.2)
Metformin (number, %)	54 (23.6)
Sodium glucose transporter-2 inhibitor (number, %)	8 (3.5)
Dipeptidyl peptidase 4 inhibitor (number, %)	46 (20.1)
Glucagon-like peptide-1 receptor agonist (number, %)	52 (22.7)
Insulin (number, %)	43 (18.8)
Mean IMT of carotid artery (mm)	1.3 ± 0.3
Maximum IMT of carotid artery (mm)	2.5 ± 0.7

Values are shown as mean ± standard deviation or number (%).

**Table 2 jcm-09-02256-t002:** Dose of each statin administered.

Statin	Dose (mg/day)	Number of Patients (%)
Atorvastatin	5	30 (13.1)
	10	24 (10.5)
	15	1 (0.4)
	20	5 (2.2)
Pitavastatin	1	3 (1.3)
	2	2 (0.9)
Pravastatin	2.5	1 (0.4)
	5	3 (1.3)
	10	15 (6.6)
Rosuvastatin	2.5	30 (13.1)
	5	63 (27.5)
	7.5	7 (3.1)
	10	24 (10.5)
	12.5	1 (0.4)
	15	7 (3.1)
	20	11 (4.8)
Simvastatin	5	2 (0.9)

**Table 3 jcm-09-02256-t003:** Simple and multiple linear regression analyses of variables potentially associated with the change in carotid mean IMT during the administration of evolocumab.

Variable	Simple Linear Regression Analysis		Multiple Linear Regression Analysis	
	Standard Coefficient	*p*-Value	Standard Coefficient	*p*-value
Age (years)	−0.125	0.06	0.027	0.69
Male sex (yes vs. no)	0.086	0.20		
Body mass index (kg/m^2^)	0.119	0.07	0.037	0.54
Systolic blood pressure (mmHg)	−0.076	0.26		
Diastolic blood pressure (mmHg)	0.065	0.33		
HDL-cholesterol (mg/dL)	−0.074	0.27		
Change in HDL-cholesterol	−0.128	0.05	−0.120	0.04
LDL-cholesterol (mg/dL)	0.048	0.47		
Change in LDL-cholesterol (mg/dL)	−0.017	0.80		
Log-triglyceride (mg/dL)	0.100	0.13		
Change in log-triglyceride (mg/dL)	−0.019	0.77		
Log-lipoprotein (a) (mg/dL)	−0.053	0.44		
Change in log-lipoprotein (a) (mg/dL)	−0.074	0.29		
Eicosapentaenoic acid to arachidonic acid ratio	−0.130	0.05	−0.039	0.52
Uric acid (mg/dL)	0.013	0.85		
Change in uric acid (mg/dL)	−0.080	0.23		
HbA1c (%)	0.116	0.08	0.056	0.38
Change in HbA1c (%)	0.012	0.86		
eGFR (mL/min/1.73 m^2^)	0.151	0.02	0.055	0.38
Change in eGFR (mL/min/1.73 m^2^)	−0.081	0.22		
Log-urine albumin/creatinine ratio (mg/gCr)	−0.065	0.35		
Hypertension (yes vs. no)	−0.073	0.27		
Diabetes mellitus (yes vs. no)	−0.046	0.49		
Coronary artery disease (yes vs. no)	−0.013	0.85		
Previous myocardial infarction (yes vs. no)	0.076	0.26		
Past or current smoking (yes vs. no)	0.031	0.64		
Statin (yes vs. no)	0.000	---		
Ezetimibe (yes vs. no)	0.019	0.77		
Probucol (yes vs. no)	0.014	0.84		
Eicosapentaenoic acid (yes vs. no)	−0.019	0.77		
Antiplatelet agent (yes vs. no)	−0.001	0.99		
Renin–angiotensin system inhibitor (yes vs. no)	−0.024	0.72		
*β*-blocker (yes vs. no)	0.007	0.92		
Calcium channel blocker (yes vs. no)	−0.039	0.56		
Diuretic (yes vs. no)	0.079	0.24		
Antihyperuricemic drug (yes vs. no)	−0.078	0.24		
Metformin (yes vs. no)	0.127	0.06	0.108	0.09
Sodium glucose transporter-2 inhibitor (yes vs. no)	0.037	0.58		
Dipeptidyl peptidase 4 inhibitor (yes vs. no)	0.054	0.42		
Glucagon-like peptide-1 receptor agonist (yes vs. no)	−0.059	0.37		
Insulin (yes vs. no)	0.037	0.58		
Mean-IMT of carotid artery (mm)	−0.481	<0.001	−0.467	<0.001
Maximum-IMT of carotid artery (mm)	−0.067	0.32		

**Table 4 jcm-09-02256-t004:** Simple and multiple linear regression analyses of variables potentially associated with the change in carotid maximum IMT during the administration of evolocumab.

Variable	Simple Linear Regression Analysis		Multiple Linear Regression Analysis	
	Standard Coefficient	*p*-Value	Standard Coefficient	*p*-Value
Age (years)	−0.046	0.49		
Male sex (yes vs. no)	0.075	0.26		
Body mass index (kg/m^2^)	0.053	0.43		
Systolic blood pressure (mmHg)	0.029	0.66		
Diastolic blood pressure (mmHg)	−0.028	0.68		
HDL-cholesterol (mg/dL)	−0.013	0.85		
Change in HDL-cholesterol	−0.173	0.009	−0.208	0.002
LDL-cholesterol (mg/dL)	−0.040	0.55		
Change in LDL-cholesterol (mg/dL)	0.041	0.53		
Log-triglyceride (mg/dL)	0.001	0.98		
Change in log-triglyceride (mg/dL)	−0.123	0.06	−0.167	0.01
Log-lipoprotein (a) (mg/dL)	−0.008	0.90		
Change in log-lipoprotein (a) (mg/dL)	0.001	0.99		
Eicosapentaenoic acid to arachidonic acid ratio	−0.094	0.16		
Uric acid (mg/dL)	−0.023	0.73		
Change in uric acid (mg/dL)	0.070	0.29		
HbA1c (%)	−0.035	0.60		
Change in HbA1c (%)	−0.081	0.23		
eGFR (mL/min/1.73 m^2^)	−0.019	0.77		
Change in eGFR (mL/min/1.73 m^2^)	−0.078	0.24		
Log-urine albumin/creatinine ratio (mg/gCr)	0.086	0.21		
Hypertension (yes vs. no)	0.110	0.10		
Diabetes mellitus (yes vs. no)	0.060	0.37		
Coronary artery disease (yes vs. no)	−0.002	0.98		
Previous myocardial infarction (yes vs. no)	−0.024	0.72		
Past or current smoking (yes vs. no)	0.025	0.71		
Statin (yes vs. no)	0.000	---		
Ezetimibe (yes vs. no)	−0.068	0.31		
Probucol (yes vs. no)	0.033	0.62		
Eicosapentaenoic acid (yes vs. no)	0.069	0.30		
Antiplatelet agent (yes vs. no)	0.146	0.03	0.121	0.06
Renin–angiotensin system inhibitor (yes vs. no)	0.133	0.04	0.095	0.15
*β*-blocker (yes vs. no)	0.097	0.14		
Calcium channel blocker (yes vs. no)	0.019	0.78		
Diuretic (yes vs. no)	0.104	0.12		
Antihyperuricemic drug (yes vs. no)	0.010	0.89		
Metformin (yes vs. no)	−0.024	0.72		
Sodium glucose transporter-2 inhibitor (yes vs. no)	0.005	0.94		
Dipeptidyl peptidase 4 inhibitor (yes vs. no)	−0.087	0.19		
Glucagon-like peptide-1 receptor agonist (yes vs. no)	0.134	0.04	0.104	0.11
Insulin (yes vs. no)	0.075	0.26		
Mean-IMT of carotid artery (mm)	0.091	0.17		
Maximum-IMT of carotid artery (mm)	−0.009	0.89		

**Table 5 jcm-09-02256-t005:** Laboratory parameters before and after 12 months of evolocumab administration.

	Baseline (*n* = 229)	12 Months (*n* = 229)	*p*-Value
Total cholesterol (mg/dL)	149.1 ± 31.7	94.3 ± 25.5	<0.001
LDL-cholesterol (mg/dL)	69.4 ± 24.1	20.8 ± 16.8	<0.001
HDL-cholesterol (mg/dL)	53.9 ± 14.0	55.4 ± 15.0	0.01
Triglyceride (mg/dL)	107 (83–151)	90 (63–125)	<0.001
Lipoprotein (a) (mg/dL)	15 (4–30)	6 (2–17)	<0.001
Uric acid (mg/dL)	4.8 ± 1.0	4.7 ± 1.0	0.05
HbA1c (%)	6.3 ± 0.9	6.5 ± 1.1	0.007
eGFR (mL/min/1.73 m^2^)	68.1 ± 17.1	66.1 ± 16.6	<0.001
Urine albumin/creatinine ratio (mg/gCr)	11.3 (5.5-37.6)	11.8 (6.5–34.0)	0.08
Alanine aminotransferase (IU/L)	21 (15–31)	21 (15–28)	0.97
Creatine phosphokinase (IU/L)	104 (77–167)	112 (76–162)	0.23

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
