# Peer review of "Effects of Evolocumab on Carotid Intima-Media Thickness and Clinical Parameters in Patients Taking a Statin"

_jcm, 2020, doi:10.3390/jcm9072256_

Round 1

Reviewer 1 Report

The authors have adequately dealt with my comments

Reviewer 2 Report

I am certainly satisfied with the changes the authors have made in their responses to me

This manuscript is a resubmission of an earlier submission. The following is a list of the peer review reports and author responses from that submission.

Round 1

Reviewer 1 Report

The original article by K. Hirai et al. is titled, “Effects of evolocumab on carotid maximum intima-media thickness and clinical parameters in patients taking a statin”.  The authors have shown some very nice work that demonstrated that evolocumab reduced the increase in carotid maximum IMT in patients taking a statin and increased HDL-cholesterol concentrations, which suggest that evolocumab is protective against carotid atherosclerosis in patients undergoing statin therapy.  Some concerns and comments follow;

Concerns:

  1. While I realize that the statistical analysis was performed using JMP 11, nevertheless, it seems extremely peculiar to me that in the Abstract section, it states in line 37 of a printed hardcopy of the manuscript that evolocumab reduced lipoprotein a from 22.9 +/- 26.3 mg/dl to 14.0 +/- 20.4 mg/dl and that this difference in the mean values was statistically significant at the p < 0.001 level. Additionally, in line 38 of a printed hardcopy of the manuscript, it was stated that evolocumab increased the serum high-density lipoprotein (HDL)-cholesterol concentration from 53.9 +/- 14.0 mg/dl to 55.4 +/- 15.0 mg/dl and that this difference in the mean values was statistically significant at the p < 0.001 level.  Using the serum HDL-cholesterol concentrations, as an example, and noting the extremely large values of the standard deviation associated with each mean value, it seems impossible that 53.9 mg/dl and 55.4 mg/dl are statistically significantly different at the p < 0.001 level.  The reported mean values are practically identical, and the s.d. for each mean value overlaps the other mean value.  This does not make scientific sense to me.

  1. This reviewer lost a little enthusiasm for the present contribution, because of statements made in lines 65 to 70 of a printed hardcopy of the manuscript. Specifically, “A recent randomized controlled trial showed that evolocumab reduced the progression of coronary atherosclerosis in statin-treated patients (8).  However, it remains uncertain whether evolocumab can reduce the progression of carotid atherosclerosis in patients taking statins.  Quite honestly, the authors have a perfect right to conduct research as they deem appropriate, but the novelty or innovation behind their study seems to be low.  It is true that Nicholls et al. examined coronary atherosclerosis, and the present authors are evaluating carotid atherosclerosis, but, again, the study somewhat lacks in originality.  This is seen in reference 8 below;

Nicholls SJ, Puri R, Anderson T, Ballantyne CM, Cho L, Kastelein JJ, et al. Effect of evolocumab on progression of coronary disease in statin-treated patients: The GLAGOV randomized clinical trial. JAMA. 2016;316(22):2373-84.

  1. This reviewer was immediately struck by the number of other medications that the cohort of patients described in Table 1 were taking besides a statin. My immediate thought was; how can the authors possibly determine that the effects they saw, as it relates to lipid concentrations and the IMT measurements, be attributed solely to evolocumab in statin-treated patients?  In fact, these are patients that are not just receiving a statin only, but they are patients receiving a statin with all kinds of other medications.  Even with multiple linear regression analysis, it would seem to this reviewer that this presents to many confounding variables to the study to form any unequivocal conclusions about the effects of evolocumab in statin-treated patients.  In fact, in lines 304 to 305 of a printed hardcopy of the manuscript, the authors state, “Therefore, we cannot exclude the possibility that other drugs, including the statins, may have affected the study results”.  I agree, and this is a major obstacle to this study, but I am thankful that they realized this possibility and acknowledged it.

  1. The authors also state in lines 284 to 287 of a printed hardcopy of the manuscript that, “These results suggest that evolocumab attenuates the progression of carotid atherosclerosis by activating reverse cholesterol transport and increasing the circulating HDL-cholesterol concentration in patients taking statins. This reviewer dos not necessarily agree with the statement, activating reverse cholesterol transport, because as far as I can determine, the authors never showed that reverse cholesterol transport was activated.  That is to say, I do not believe that there were any measurements performed to prove that reverse cholesterol transport was activated.  Are the authors’ just assuming that because HDL was increased in their study (which is questionable; see comment/concern #1) and it is known that evolocumab can increase the activity of the ABCA1 transporter and stimulate cholesterol efflux to HDL, that reverse cholesterol transport was somehow activated?  It might be more appropriate to say, “these findings would tend to support the premise that reverse cholesterol transport was most likely activated due to the use of evolocumab in our study, but an increase in the rate of reverse cholesterol transport was not specifically quantified in the present study”.

Reviewer 2 Report

The topic of this study is interesting and I would be happy to see that the addition of a PCSK9 inhibitor on statin attenuates the progression of carotid atherosclerosis.

However, there are major limitations which affect the validity of this study.

Main limitations

1) The major limitations are the retrospective design of the study and the luck of control group

2) Another concern is the methodology of assessing carotid IMT. The authors measured manually the maximum IMT of cCA, carotid bulb, and internal CA. My feeling is that a more appropriate measurement is the average of all mean IMT values obtained from left and right cCA, carotid bulb, and internal CA by automated analysis techniques of B‐mode ultrasound imaging. Automated edge‐detection is usually preferred over manual analysis since it has been shown to be more accurate.

Minor limitations

1) The proportion of FH patients in the study is very low (<1%). In most studies which include patients taking PCSK9-inhibitors on top of statin treatment the prevalence of FH is much higher

2) What was the reproducibility of carotid IMT measurements?

3) What was the statin dose?

4) The authors state that the patients were prescribed evolocumab on top of statins treatment because they were high-risk patients due to the presence of carotid artery plaques. This is not a clear indication for PCSK9 inhibitors. Do the health authorities approve the prescription of a PCSK9 inhibitor for carotid plaques causing <50% stenosis?

5) How the authors explain the mild diabetogenic effect of evolocumab?